# Detection of Chitin Synthase Mutations in Lufenuron-Resistant *Spodoptera frugiperda* in China

**DOI:** 10.3390/insects13100963

**Published:** 2022-10-20

**Authors:** Sheng-Lan Lv, Zheng-Yi Xu, Ming-Jian Li, Amosi Leonard Mbuji, Meng Gu, Lei Zhang, Xi-Wu Gao

**Affiliations:** Department of Entomology, China Agricultural University, Beijing 100193, China

**Keywords:** *Spodoptera frugiperda*, lufenuron, chitin synthase, resistance

## Abstract

**Simple Summary:**

Lufenuron is one of the main insecticides for controlling lepidopteran pests, and is especially suitable for controlling pests that are resistant to pyrethroids and organophosphorus pesticides. Resistance to lufenuron is a serious obstacle to effective pest control. A major mechanism of lufenuron resistance is due to mutations in chitin synthase, the target of the toxic effects of benzoylurea (BPUs), according to a reported mutational analysis, I1040 (I1042 in *Plutella xylostella*) in the SfCHSA gene. The mutation of the site can significantly improve the resistance of insect larvae to BPU insecticides. Secondly, the expression level of insecticide targets in insects is also an important factor affecting insect resistance. In this study, we firstly identified the number of chitin synthase genes and their corresponding names in *Spodoptera frugiperda*, and detected the mutation sites of the SfCHSA gene. The expression level of chitin synthase gene was then analyzed. Our results showed that no mutation was found in the SfCHSA gene in the currently low-resistant field populations in China, and the expression of chitin synthase was affected by lufenuron. The information obtained from this study is valuable for the use of lufenuron in the field.

**Abstract:**

*Spodoptera frugiperda* (J. E. Smith), is commonly known as fall armyworm, native to tropical and subtropical regions of America, is an important migratory agricultural pest. It is important to understand the resistance and internal mechanism of action of *S. frugiperda* against lufenuron in China. Lufenuron is one of the main insecticides recommended for field use in China and has a broad prospect in the future. We conducted a bioassay using the diet-overlay method and found that the current *S. frugiperda* in China are still at a low level of resistance to lufenuron. Secondly, we examined whether the mutation I1040M (I1042M in Plutella xylostella), associated with lufenuron resistance, was produced in the field. And then we tested the expression of chitin synthase SfCHSA and SfCHSB in different tissues, and the changes of these two genes after lufenuron induction. The results showed that there is still no mutation generation in China and there is a significant change in the expression of SfCHSA under the effect of lufenuron. In conclusion, our study suggests that field *S. frugiperda* populations in 2019 and 2020 were less resistant to lufenuron. In fall armyworm, chitin synthases included SfCHSA and SfCHSB genes, and after induction treatment with lufenuron, the expression of the SfCHSA gene was significantly increased. In SfCHSA, no mutation has been detected in the site associated with lufenuron resistance. Secondly, in *S. frugiperda* larvae, the SfCHSA gene was the highest in the head of the larvae, followed by the integument; while the SfCHSB gene was mainly concentrated in the midgut. Therefore, we believe that the SfCHSA gene plays a greater role in the resistance of *S. frugiperda* to lufenuron than the SfCHSB gene. It is worth noting that understanding the level of resistance to lufenuron in China, the main mechanism of action of lufenuron on larvae, and the mechanism of resistance to lufenuron in *S. frugiperda* will help in crop protection as well as in extending the life span of this insecticide.

## 1. Introduction

*Spodoptera frugiperda* (J. E. Smith), commonly known as fall armyworm, native to tropical and subtropical regions of America, is an important migratory agricultural pest [1,2,3,4]. In the period of serious harm, it was found to have invaded Yunnan province, being believed to have come from Myanmar, then quickly moved east and north by migrating to most provinces and cities in China [5,6,7]. In Africa and Asia, it mainly harms maize [4,8,9]; the larvae like to damage the core leaf clusters and corn grains at the filling stage. In the period of serious harm, the corn even had no grain [4,10,11,12]. It is also reported that it harms sorghum, sugar cane, wheat, potato, and other crops in China [13,14,15,16].

Chitin is a polysaccharide monomolecular polymer composed of N-acetyl glucosamine. It is an important component of invertebrate animal shell and fungal algae cell walls, and plays an important role in individual life activities [17,18,19,20]. In insects, chitin is an important component of the epidermis, trachea, and perineum of the midgut [21]. Chitin synthase plays a key role in insect growth and is development mainly by acting on chitin metabolism. At present, chitin synthase in insects is mainly encoded by two genes, chitin synthase 1 (CHSA) and chitin synthase 2 (CHSB), which exist on the same chromosome and may have evolved from a gene copy of the same ancestor [21,22,23,24]. Point mutations are one of the important causes of enhanced insecticide resistance in insects. For example, a mutation in the ryanodine receptor (I4734M) resulted in more than 200-fold enhanced resistance to chlorantraniliprole in *S. frugiperda* [25]. Studies have shown that the replacement of amino acid I1042M in CHSA is highly correlated with resistance of diamondback moth to several benzoylureas (BPUs), such as diflubenzuron, triflumuron, lufenuron, and flucycloxuron [26]. In addition, this mutation was introduced in *Drosophila melanogaster* using CRISPR/Cas9 combined with homology-directed repair (HDR) genome modification, which showed resistance to different chemical classes of insect-growth regulators [26]. Other than, in the study of Ishaaya and Casida [27], the chitinase activity of a house fly was increased when fed with diflubenzuron-treated diet; Zhang and Zhu [28] reported that after treatment with diflubenzuron, *Anopheles quadrimaculatus*’ CHS1 expression was up-regulated. Based on the published studies, we believe that there is a link between BPUs and CHS.

Insect-growth regulators (IGRs) can inhibit insect physiological development, such as by inhibiting molting, inhibiting the formation of new epidermis, and inhibiting feeding, which lead to the death of pests. Its mechanism of action is different from traditional insecticides that act on the nervous system. Low toxicity, less pollution, and little impact on natural enemies and beneficial organisms contribute to the development of sustainable agriculture, are conducive to the production of pollution-free green food, and are beneficial to human health. Therefore, these are known as “third generation pesticides”, “pesticides of the 21st century”, “bio-regulators”, and “novel materials for insect control” [29]. Because of their unique mechanism of action and advantages in sustainable agriculture, IGRs are widely used in pest control around the world [30]. Lufenuron belongs to benzoylureas (BPUs), an insect-growth regulator, which mainly inhibits the synthesis of chitin and makes insects unable to molt or pupate and thus finally die. Its main mode of action is stomach toxicity, which is effective against a variety of pests. At low concentrations, it can prolong the developmental period of larvae, and reduce the pupation rate, eclosion rate, and egg-laying rate of insects. When the concentration is high, it can also directly kill eggs and larvae [31,32,33,34]. Therefore, such insecticides can be used not only to control lepidoptera larvae on cotton, corn, vegetables, and fruit trees [35], but also to control other insects, such as *Phyxia scabiei Hopk* and *Apolygus lucorum* [34,36].

Up to now, the resistance level and target-gene mutation frequency of chitin synthase in *S. frugiperda* in China have remained unknown. How many chitin synthase genes are present in *S. frugiperda* and which gene plays a huge role in insecticide resistance has also not been reported. Measuring the potential risk of resistance evolution and resistance mechanisms could help extend the life of effective insecticides for pest control.

In this paper, we investigated the resistance of eight *S. frugiperda* strains from China against lufenuron from 2019 to 2020. To determine the risk of resistance of this pest to lufenuron, we studied the gene of chitin synthase in *S. frugiperda*, which is important for us to further understand the resistance mechanism of this insect to lufenuron. In addition, we also detected the mutation at position 1040 (1042 in *Plutella xylostella*) of the chitin synthase (SfCHSA) gene in a field strain of *S. frugiperda* at different locations in different years. Studying which chitin synthase gene is more likely to be involved in insecticide resistance provides evidence to minimize the risk of *S. frugiperda*’s resistance to lufenuron.

## 2. Materials and Methods

### 2.1. Insect Populations

In 2019 and 2020, we collected a total of 16 strains of *S. frugiperda* from 8 different locations in China. The SUS strain is the Wuhan field population in 2019 that has been maintained in the laboratory without insecticide selection for more than 1 year. All insects were collected during the larval stage and reared in an insectary at 27 ± 1 °C, 70–80% RH and a 16:8 (L:D) h dark photoperiod.

Larvae from all strains were raised on an artificial diet based on corn flour, brewer’s yeast, soybean meal, and yeast powder [37]. The adults were fed with 10% honey solution and maintained in a plastic bucket (20 cm diameter × 30 cm height) covered internally with a gauze (oviposition substrate). 

### 2.2. Chemicals

Lufenuron of 96% purity was obtained from Swiss Syngenta crop protection Co., Ltd. (Shanghai, China). Acetone and Triton X-100 were purchased from Sinopharm Chemical Reagent Co., Ltd. (Shanghai, China).

### 2.3. Bioassay

A modification on the method described by Bolzan et al. was performed [38]. The specific steps were: dissolve the lufenuron powder with acetone, configure it as a solution of higher concentrations, and then continuously dilute it with 0.1% Triton aqueous solution into 5–7 gradients. In a 24-well culture plate, 1 mL of liquid artificial feed cooled to about 60 °C was added to each well separately and allowed to cool for natural curing. A total of 50 μL of different concentrations were then added to each well and evenly coated on the surface of the feed, left to dry at room temperature, and then a second instar larva was placed into each well. Each concentration was replicated four times, with 12 larvae per replicate, and 0.1% Triton aqueous solution was used as the control. The results were observed after 96 h of treatment in a suitable environment. If the larvae did not respond to touch with a brush, or they showed obvious symptoms of poisoning (malformation, twitching, cessation of feeding, etc.), they were considered to be dead.

### 2.4. Bioinformatics Analysis

The NCBI online tool BLAST (http://blast.ncbi.nlm.nih.gov/Blast.cgi, accessed on 1 May 2020) was used to search for the homology of proteins; the MEGA-X neighbor-joining method was used to analyze the phylogenetic relationship.

### 2.5. Sequencing Partial SfCHSA

DNA from 4 populations of insect individuals of different years was extracted using tissue/cell genomic DNA isolation kits (solutions) (BioTeKe Corporation, Beijing, China), following the manufacturer’s instructions. Two pairs of primers (Table 1) were designed using DNAMAN 8.0 (LynnonBiosoft, San Ramon, CA, USA) to amplify the fragments of SfCHSA where mutations have been reported. PCRs were performed in a final volume of 25 μL with 12.5 μL 2× Taq Master Mix for PAGE (Dye Plus) (Vazyme, Nanjing, China), 2 μL of forward and reverse primers (see Table 1), and 1 μL DNA and 7.5 μL nuclease-free water. The PCR primer cycle consisted of an initial denaturation phase of 3 min at 94 °C and then 35 cycles including denaturation at 94 °C for 15 s, annealing at 59 °C for 5 s, and then extension at 72 °C for 15 s. PCR ended with a final extension of 5 min at 72 °C. The pyrosequencing reaction was carried out by SinoGenoMax (Beijing, China) using the sequencing primers listed in Table 1. Analyze the sequencing results to examine the mutations in the field population of the *S. frugiperda.*

### 2.6. Gene Expression of SfCHS in Different Tissues

Fifteen to twenty well-grown and uniform larvae from the second day of the sixth instar of *S. frugiperda* were collected and dissected in normal saline with stereomicroscopy. The tissues in these insects are intact and easy to dissect. Head, integument, midgut, and fat body of *S. frugiperda* were isolated. After the material was collected, it was immediately put into liquid nitrogen to cool and protect it, then stored in a refrigerator at −80 °C for later use.

Total RNA was extracted from the collected samples with Trizol reagent. Reverse transcription was performed on 1 μg of each RNA sample in a 20 μL reaction with Hiscipt^®^ II QRT SuperMix for qPCR (+gDNA wiper) using the protocol supplied with the kit. Real-time quantitative PCR (RT-qPCR) was conducted using ChamQ Universal SYBR qPCR Master Mix (Vazyme, Nanjing, China). Ribosomal protein L18 was used as a reference gene [39]. RT-qPCR was carried out in a reaction of 20 μL containing 1.0 μL of the complementary DNA (cDNA), 10 μL of 2× ChamQ Universal SYBR qPCR Master Mix, 0.4 μL each of the forward and reverse primers (10 μmol L^−1^), and 8.2 μL nuclease-free water. Primers for *S. frugiperda* CHSA, CHSB, and L18 genes (Table 1) were designed using DNAMAN 8.0 software (LynnonBiosoft, San Ramon, CA, USA). Thermal cycling conditions were: 50 °C for 2 min, 95 °C for 30 s, 40 cycles of 95 °C for 15 s, 60 °C for 34 s, 72 °C for 30 s. After the cycling protocol, a melting-curve analysis from 60 °C to 95 °C was applied to all reactions to verify a single PCR product. The amplification efficiency was estimated using the equation E = 10 °C/slope, where the slope was derived from the plot of cycle threshold (Ct) value versus six serially diluted template concentrations. The experiment was set up with three biological replicates and three technical replicates were performed.

### 2.7. Effects of Lufenuron on the Expression of SfCHS Gene

The sub-lethal concentration (LD_30_) [40] of the Hainan *S. frugiperda* strain to lufenuron was 0.006 µg a.i. cm^−2^ in 2019. Three groups of biological replicates were set up with 24 third-instar *S. frugiperda* larvae in each group, and 0.1% Triton water was used as the control. After feeding for 96 h, the fall armyworms in the control group and treatment group were collected, and whole *S. frugiperda* RNA was extracted. The first strand of cDNA was synthesized and detected by real-time fluorescence quantitative PCR. The specific methods and procedures were as in Section 2.6.

### 2.8. Statistical Analyses

LD_50_ values, 95% fiducial limits (FL), slope, SE, confidence-interval overlap, and ratio tests of median lethal concentration values were calculated using PoloPlus software (LeOra Software Inc., CA, USA). The relative resistance ratio (RR) was calculated by dividing the LD_50_ value of the field population divided by that of a susceptible population. The test data were expressed as “mean ± standard error (SE)”. One-way ANOVA was performed on the chitin-synthase-expression test data using SPSS17.0 software, and the expression data before and after treatment were analyzed for significance using Student’s *t*-test.

## 3. Results

### 3.1. Bioassays

Diet-overlay bioassays were conducted in 24-well acrylic plates to evaluate resistance of *S. frugiperda* to lufenuron. We collected eight population strains from eight counties in China: Haikou, Guangzhou, Kunming, Jiujiang, Shaoyang, Wuhan, Mianyang, and Qinzhou. The SUS strain was sensitive to lufenuron with a LD_50_ value of only 0.010 μg a.i. cm^−2^. Compared with the SUS strain, all strains showed a sensitive level of resistance (RR < 5). In 2019, the eight populations had an RR of 1.2, 2.3, 2.1, 2.3, 3.1, 2.4, 1.8, and 2.8, respectively. In 2020, the eight populations had an RR of 3.6, 2.2, 2.0, 2.4, 2.1, 3.4, 4.0, and 2.7 (Table 2).

### 3.2. Homology Comparison and Molecular Phylogenetic Analysis of SfCHSA and SfCHSB Genes

The SfCHSA and SfCHSB genes were compared with the NCBI database (http://ncbi.nlm.nih.gov, accessed on 1 May 2022), The neighbor-joining method in MEGA-X was then used for molecular phylogenetic analysis, and the molecular evolutionary tree of the insect chitin synthase gene was analyzed (as shown in Figure 1). It can be seen from the phylogenetic tree that the chitin synthase gene is divided into two branches, namely, chitin synthase 1 (CHSA) and chitin synthase 2 (CHSB). Chitin synthetase 1 (SfCHSA, SeCHS1, HzCHS1, HaCHS1, MsCHS1, and PxCHS1) forms a subclade in all lepidoptera. SfCHSA was associated with *Spodoptera exigua* (Hubner), *Helicoverpa armigera* (Hubner), *Helicoverpa zea*, and *Manduca sexta*; they are closely related to *P. xylostella* chitin synthase 1 (PxCHSA), which is consistent with the traditional classification.

Note: The chitin synthase gene sequences of the selected biological species include *Anopheles gambiae* (AgCHSA, XP_321336; AgCHSB, AAF34699.2), *Apis mellifera* (AmCHSA, XP_395677; AmCHSB, XP_001121152), *Drosophila melanogaster* (DmCHSA, NP524233; DmCHSB, NP001137997), *Lucilia cuprina* (LcCHSA, AAG09712; LcCHSB, XM_046953320.1), *Manduca sexta* (MsCHSA, AAL38051; MsCHSB, AAX20091), *P. xylostella* (PxCHSA, BAF47974; PxCHSB, XM_038114800.2), *Spodoptera exigua* (SeCHSA, AAZ03545; SeCHSB, ABI96087), *Tribolium castaneum* (TcCHSA, AAQ55059; TcCHSB, AAQ55061), *Helicoverpa zea* (HzCHSA, ADX66429; HzCHSB, XM_047186417), *Helicoverpa armigera* (HaCHSA, AKJ54482.1; HaCHSB, KT004448).

### 3.3. Detection of Mutations of CHSA in the S. frugiperda Population

First, we sequenced the PCR product derived from the individual DNA of selected strains. The strains we selected were those with different resistance to lufenuron in 2019 and 2020, including 33 insects from Hainan, 22 insects from Hunan, 24 insects from Fujian, and 22 insects from Jiangxi in 2019 and 17 insects from Hubei, 27 insects from Yunnan, 45 insects from Guangxi, and 30 insects from Sichuan in 2020. The determined nucleotide sequence of the PCR product (200 bp) was found to be 99% identical to the counterpart from the sensitive strain. Thus, we confirmed that the PCR procedure indeed amplified the specific product correctly. The sequencing results were analyzed to detect mutations associated with BPU insecticide resistance. The results showed that the amino acid fragments amplified by *S. frugiperda* field did not have I1040M/I1040F mutations (Figure 2).

### 3.4. Expression of SfCHS Gene in Different Tissues

SfCHSA and SfCHSB gene expression levels were detected in different tissues of the sixth instar larvae of *S. frugiperda*. The results showed (Figure 3) that SfCHSA gene expression level was the highest in the head, followed by the integument and fat body, and the lowest in the midgut. SfCHSB gene expression level was the highest in the midgut. The expression levels in the integument, head and fat body were extremely low.

### 3.5. Effect of Lufenuron on the Expression of Chitin Synthase Genes (SfCHSA and SfCHSB) in S. frugiperda

The third instar larvae of the Hainan population in 2019 were treated with lufenuron at a concentration of 0.006 μg a.i. cm^−2^, and the gene-expression level of chitin synthase (SfCHSA and SfCHSB) in the larvae was measured by RT-qPCR at 96 h after treatment. The results showed that after 96 h of treatment with lufenuron, the expression level of chitin synthase gene in *S. frugiperda* larvae was significantly higher than that control larvae (Figure 4). The expression of SfCHSA increased significantly (*p* < 0.01), but the expression of SfCHSB did not change significantly.

## 4. Discussions

In this study, we characterized the resistance of field collected population of *S. frugiperda* to lufenuron from eight places. In 2019, Hainan, Guangdong, Guangxi, Hubei, Hunan, Jiangxi, Sichuan, and Yunnan were all sensitive strains with resistance multiples of 1.2–3.1; in 2020, the resistance multiple of all populations was between 2.0–4.0, and they were also sensitive populations. In Nascimento’s report [42], the LD_50_ value of lufenuron against *S. frugiperda*, a sensitive population propagated in the laboratory for 10 years, was 0.0045 μg a.i. cm^−2^, as measured by the diet-overlay method. The difference between this value and our sensitive value was small, which proves that the invasion of *S. frugiperda* is very sensitive to lufenuron. Therefore, when controlling *S. frugiperda* in the field, lufenuron can continue to be used.

Douris et al. [26] identified a mutation (I1042M) in the chitin synthase gene 1 (PxCHSA) in *P. xylostella* that had developed resistance to BPUs, and that *Drosophila* also developed resistance to these insecticides when this mutation was introduced into *Drosophila* using the CRISPR/cas9 method. In addition, the study also revealed the fact that another mutation at this site (I1042F) was associated with resistance to etoxazole in mites, and again after verification in *Drosophila*, this mutation was also found to be associated with BPUs. Therefore, either I1042M or I1042F can cause insects to develop resistance to BPU-like insecticides. Furthermore, we also found the same type of mutation (I1017F) within the chitin synthase gene 1 (SfCHSA) at this site in a field population of *Frankliniella occidentalis* that had developed resistance to lufenuron [43]. In the genome of *S. frugiperda*, we obtained two different chitin synthase genes, SfCHSA and SfCHSB, and detected resistance-related mutation sites in SfCHSA. Our results showed that we were unable to find mutations at the I1040 (I1042 in *P. xylostella*) site in the field strain, which is consistent with our bioassay results.

The formation of the epidermis is crucial to the growth and development of insects throughout the molting cycle of larval development [44]. Insects synthesize and degrade chitin with a certain pattern during their development to ensure the completion of molting and the regeneration of the exoskeleton, trachea, etc. The synthesis of chitin in the organism is catalyzed by a series of enzymes. It was found that the majority of insects contain two different chitin synthase genes, CHS1 and CHS2. The CHS1 gene is mainly expressed in insect epidermal and tracheal tissues, while the CHS2 gene is mainly expressed in midgut cells [45,46].The expression of SfCHSA and SfCHSB genes was examined by real-time PCR in different tissues of the *S. frugiperda*, and it was found that SfCHSA genes were mainly expressed in the head and integument, while the expression was low in the midgut and fat body, and the results were consistent with previous studies in the *Locusta migratoria manilensis* [47], *M. sexta* [24], and *T. castaneum* [31]. Besides this, the SfCHSB gene is mainly expressed in the midgut of the *S. frugiperda*, with little expression in the head, integument, and fat body. These results are in agreement with Wang’s studies in *Mythimna separata* and Bolognesi’s studies in *S. frugiperda* [48,49].

Benzoylurea insect-growth regulators mainly interfere with the formation and metabolism of epidermal chitin in insects, and cause damage to their growth and development by affecting their normal molt, but their exact mechanism of action is not yet known. Lufenuron is an inhibitor of benzoylurea chitin synthesis, which is highly toxic to lepidopteran larvae and has a strong ovicidal effect [50]. It was found that after treating the third instar larvae of *S. frugiperda* with a sub-lethal concentration of lufenuron (0.006 μg a.i. cm^−2^), the expression of the SfCHSA gene increased significantly in the test insects 96 h after the insecticide treatment, and the expression of the SfCHSB gene did not change significantly. Therefore, we believe that the insecticidal effect of larvae of lufenuron is most closely related to SfCHSA gene, but the role of SfCHSB gene in the insecticidal process cannot be denied for this reason.

At present, the resistance of *S. frugiperda* to lufenuron is low in China, and no mutation related to lufenuron resistance has been detected. We believe that lufenuron is still a highly recommended insecticide for the control of *S. frugiperda* in the field. Secondly, through the study and analysis of the chitin synthase gene, we suggest that the SfCHSA gene plays an important role in the development of resistance to lufenuron in the *S. frugiperda*, but the specific mechanism needs to be studied next.

## Figures and Tables

**Figure 1 insects-13-00963-f001:**
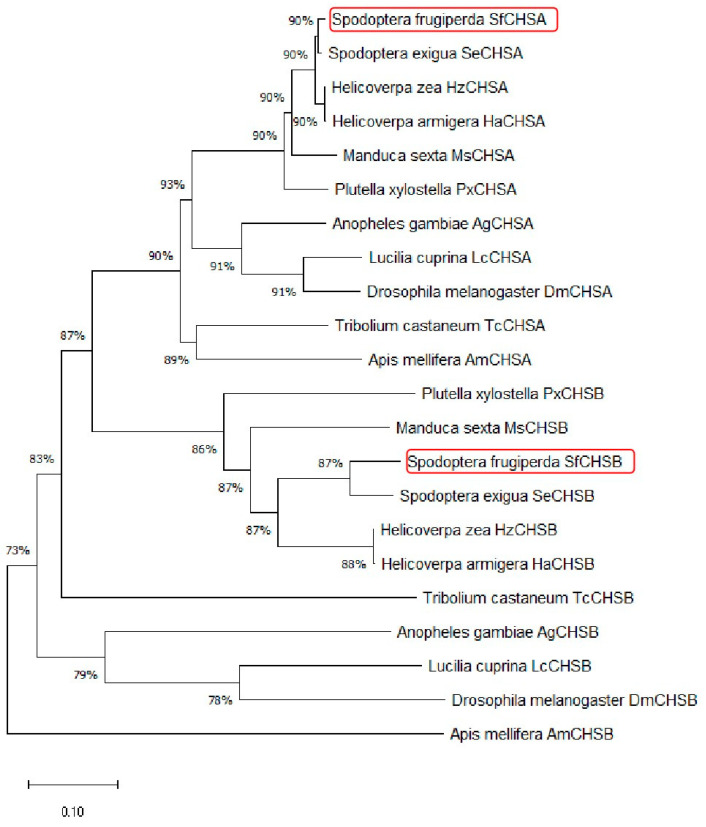
Phylogenetic tree based on the amino acid sequences of insect chitin synthases. The red border indicates the chitin synthase of *S. frugiperda*.

**Figure 2 insects-13-00963-f002:**
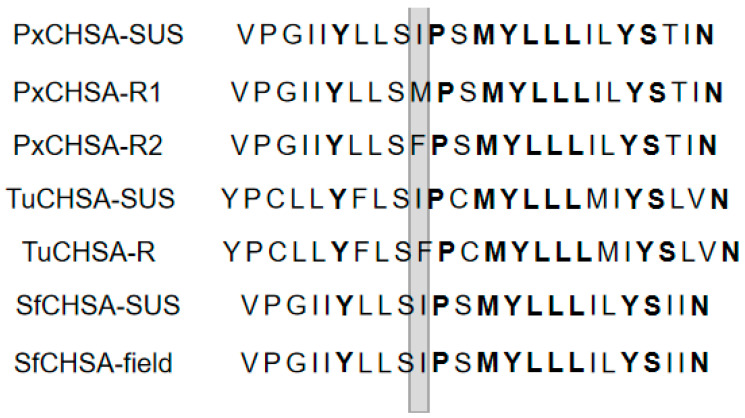
Aligned amino acid sequences of helix 5 in the C-terminal 5 transmembrane segments of CHSA. The positions of the I1017F substitution in etoxazole-resistant *Tetranychus urticae* [41] and I1042M/F in BPU insecticide-resistant *P. xylostella* [26] are indicated in gray. Conserved amino acid residues are shown in bold letters. PxCHSA-SUS, *P. xylostella* CHSA of the susceptible strain BCS-S (DDBJ/EMBL/GenBank accession number API61825); PxCHSA-R1, *P. xylostella* CHSA of the resistant strain [26]; PxCHSA-R2, *P. xylostella* CHSA of the resistant strain (China) [33]; TuCHSA-SUS, *T. urticae* CHSA of the etoxazole susceptible strain GSS (AFG28413); TuCHSA-R, *T. urticae* CHSA of the etoxazole-resistant strain EtoxR (AFG28419); SfCHSA-SUS, *S. frugiperda* CHSA of the susceptible strain; SfCHSA-field, *S. frugiperda* CHSA of field strains.

**Figure 3 insects-13-00963-f003:**
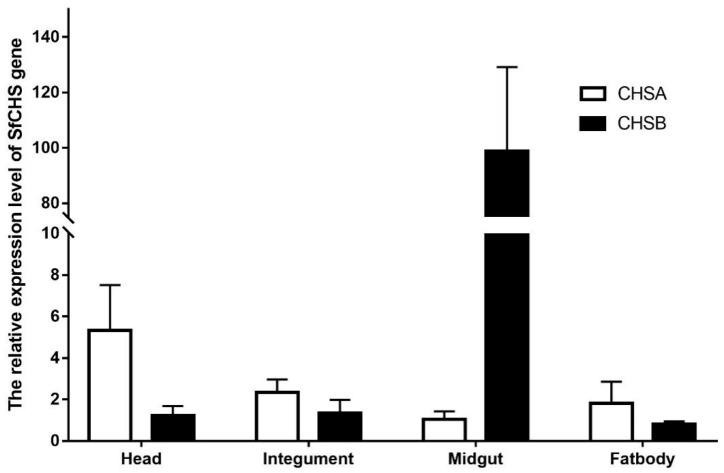
Relative expression levels of chitin synthase genes (SfCHSA and SfCHSB) in different tissues of *Spodoptera frugiperda.* Note: the lowest expression is ascribed an arbitrary value of 1; expression levels in integument, midgut, head, and fat body were detected by qPCR.

**Figure 4 insects-13-00963-f004:**
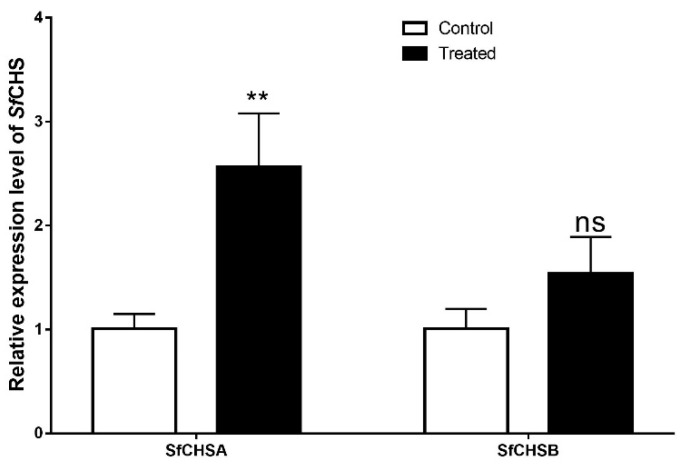
Relative expression levels of SfCHSA and SfCHSB after exposure to lufenuron. Note: ** on the column indicates a significant difference between the treatment group and the control group (*p* < 0.01), ns means not significantly different (*p* > 0.05).

**Table 1 insects-13-00963-t001:** List of primers used for pyrosequencing and RT-qPCR.

Primers	Sequence	
I1040M-F	5′-ATCTCCTTCGGCTATATTCTTGAT-3′	PCR
I1040M-R	5′-CTTCGTCTTCTTAACTGCCACTTC-3′	
L18-F	5′-CGTATCAACCGACCTCCACT-3′	RT-qPCR
L18-R	5′-AGGCACCTTGTAGAGCCTCA-3′	
CHSA-F	5′-TCCTTATGTTGGTGGGTGCC-3	
CHSA-R	5′-GTACCGACGATCACAGCCAT-3′	
CHSB-F	5′-ATCCAGTTCACCGCCATGTT-3′	
CHSB-R	5′-CCAAGTCGTCGGTGTTCAGA-3′	

**Table 2 insects-13-00963-t002:** Resistance levels of field population of *S. frugiperda* to lufenuron by diet-overlay bioassay.

Insecticide	Strain	Coordinate	Year	Slope ± SE	LD_50_ (95% CL) (μg a.i. cm^−2^)	Df (χ^2^)	RR
Lufenuron	SUS	114.8° E 30.85° N	2019	1.226 ± 0.171	0.010 (0.007–0.014)	18 (14.48)	1.0
	Hainan	110.28° E 20.02° N	2019	1.376 ± 0.229	0.012 (0.007–0.016)	14 (12.90)	1.2
			2020	2.216 ± 0.311	0.036 (0.030–0.045)	18 (5.922)	3.6
	Guangdong	113.22° E 23.4° N	2019	2.028 ± 0.298	0.023 (0.017–0.029)	14 (7.528)	2.3
			2020	1.017 ± 0.170	0.022 (0.017–0.030)	21 (2.965)	2.2
	Guangxi	103.14° E 24.40° N	2019	2.327 ± 0.321	0.021 (0.017–0.027)	14 (5.599)	2.1
			2020	2.579 ± 0.309	0.020 (0.017–0.024)	18 (16.78)	2.0
	Hubei	114.8° E 30.85° N	2019	1.648 ± 0.218	0.023 (0.017–0.031)	18 (11.29)	2.3
			2020	1.400 ± 0.229	0.024 (0.016–0.032)	18 (6.196)	2.4
	Hunan	111.73° E 27.25° N	2019	1.803 ± 0.266	0.031 (0.022–0.039)	14 (10.25)	3.1
			2020	1.599 ± 0.240	0.021 (0.014–0.027)	18 (8.065)	2.1
	Jiangxi	115.8° E 29.02° N	2019	2.263 ± 0.299	0.024 (0.019–0.029)	18 (9.056)	2.4
			2020	1.047 ± 0.173	0.034 (0.023–0.050)	22 (8.807)	3.4
	Sichuan	105.06° E 31.1° N	2019	2.883 ± 0.348	0.018 (0.015–0.022)	18 (14.06)	1.8
			2020	1.092 ± 0.216	0.040 (0.026–0.058)	18 (12.47)	4.0
	Yunnan	103.15° E 24.76° N	2019	2.983 ± 0.417	0.028 (0.022–0.033)	14 (8.952)	2.8
			2020	1.347 ± 0.223	0.027 (0.018–0.037)	18 (9.651)	2.7

LD_50_: Lethal dose resulting in 50% dead or severely injured; 95% CL: confidence limits at 95. RR: Resistance Ratio = LD_50_ of field population/LD_50_ of sensitive strain. Df: Degrees of freedom.

## Data Availability

The data presented in the study are available in the article.

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
