# Peer review of "Detection of Chitin Synthase Mutations in Lufenuron-Resistant Spodoptera frugiperda in China"

_insects, 2022, doi:10.3390/insects13100963_

Round 1

Reviewer 1 Report

The manuscript by Sheng-Lan Lv and his colleagues investigated chitin synthase mutations in lufenuron resistant Spodoptera frugiperda in China. The manuscript is written scientifically, and the data is sufficient to support its main conclusion. It meets in the present form the criteria to be published in the journal. However, several points should be addressed prior to publication. 

Line 4, who was correspondent author, please revise.

Line 15, CHSA? SfCHSA?

Line 56, please delete “”.

Line 65, “S. frugiperda” should be in italic, similarly hereinafter.

Line 81, please add “.” after year.

Line 105, Why sequencing after third instar S. frugiperda was treated with sublethal concentration of lufenuron (LC30)? Please clarify.

Line 107, in “LC30”, “30” should be subscripted.

Line 120, “mL” should be “µL”.

Line 128, why use larva of the middle sixth instar? Please clarify.

Line 132, the format of “” is not consistent with that of other “” in the article, please revise.

Line 134, the format of “u” is not consistent with that of other “u” in the article, pleased revise, similarly hereinafter.

Line 150, please delete “)”.

Line 150, how old larvae were selected?

Line 157, does “SE” need to be in italic?

Line 170, LC50? LD50?

In table 2, Df means degree of freedom? Why is this value so large? In “χ2”, “2” should be subscripted.

Line 212, “SCHSB” should be “SfCHSB”.

Line 273, please delete “;”.

Line 274, “” should be “,”.

Line 278, LC30/LD30 was 0.020 ug a.i.cm-2, which strain was used?

Line 280, it's not just the stomach poison toxicity, please revise.

In figure 5, “CHSA/CHSB” should be “SfCHSA/SfCHSB”.

Line 302, “Crispr” should be “CRISPR”.

Line 326, “,” should be “.”.

Line 333, please delete “the” and “larvae”.

In figure 4, why SfCHSB gene expression level was high in midgut?

Reviewer 2 Report

This study targets an important agricultural pest, Spodoptera frugiperda. Through the bioassay, authors found that S. frugiperda in China are still at a low level of resistance to lufenuron. In addition, according to mutation positions in other studies, the results showed that there is still no mutation generation in China. This paper provides a certain direction for the research of S. frugiperda resistance mechanism on lufenuron. And I quite approve of this work, which still have some questions by the authors to modify, detailed modification suggestions are as follows:

Abstract:

1.      Line 8: The latin words “Spodoptera frugiperda” should be consistent with the standard of the journal, also including Line 65-66 and some Chinese marks. Please check the whole text carefully.

2.      Line 14: It is not mentioned in the abstract of which mutations related to resistance are not appeared in Spodoptera frugiperda.

Introduction:

3.      Line 45: The first time BPU is abbreviated, the full name should be described. All other abbreviations should be checked carefully.

4.      Line 50: Suggest changing “inhibit feeding” to “inhibiting feeding”, be consistent with the whole sentence.

5.      Line 61: Authors introduce the toxicity of lufenuron to lepidopteran pests detailed, its better to add some descriptions of insecticidal effects of lufenuron on other order pests, due to the mutation is also mentioned in Drosophila melanogaster.

Materials and methods:

6.      Line 81: Missing a period.

7.      Line 120: The PCR final volume is 25 mL or 25 μL, please check it, and all the units of volume should be μL, not uL, such as Line134-140.

8.      Line 125: We can know from 2.7, author also put RT-qPCR primer into table 1, so the table 1 title should be modified to summarize the information of all primers.

Results:

9.      Line 170: Table 2 is LD50, while SUS strain was sensitive to lufenuron with a LC50 value……, please confirm the bioassay method is LD50 or LC50 and consistent with the full text.

10.  Line 240: Table 3 title shows eight strains of “P. xylostella”in different places, while not “S. frugiperda”? Please check it.

11.  Line 256: Figure 3 is the phylogenetic analysis of sfCHSA and sfCHSB gene, while the title of 3.4 just mentioned sfCHSA. Why do some species on the evolutionary tree use both CHSA and CHSB, while other species only use CHSA?

12.  Is there an appropriate ratio for field application of lufenuron, which can be suggested to be more helpful for field application since excessive pesticide use can increase pest resistance.

13.  From the results of bioassay, the resistance of S. frugiperda to lufenuron is low in China, are there other pesticides to S. frugiperda with strong resistance to study the mutation site? Resistance to other pesticides could be described in the introduction.

Discussion:

14.  In 2019 and 2020, authors collected eight strains of S. frugiperda. Whether a total of 8 strains were collected in two years or 8 strains were collected every year, which is not detailed in the experimental method.

References:

15.  There are some mini questions, such as italic text of latin, Chinese punctuation marks and page number labeling is not uniform. Including reference 13, 17, 19, 20, 21, 23, 24 and so on. Please check them all carefully.

Reviewer 3 Report

Dear authors, congratulations by the study where determined that the S. frugiperda on the time actual not show resistance to lufenuron

Reviewer 4 Report

Comments to the manuscript insects-1904599 intended as a research article in Insects entitled “Detection of chitin synthase mutations in lufenuron resistant Spodoptera frugiperda in China” by Lv et al. and six coauthors. They present an interesting manuscript on lufenuron resistance in the fall armyworm.

A project with a very nice bioassay design. The methods used, the number of replicates and the analyses performed seem appropriate and solid, but with a complete lack of information about the genome sequencing and isolation of SfCHSA and SfCHSB.

Figure 1 is not needed and should be replaced with a Supplementary Table with GPS coordinates or the GPS coordinates can be added to Table 2 or they could be mention as text in Section 2.1.

Ln95; please replace “mother liquor” with a more scientific expression.

Ln107; it does not make sense to treat with insecticide before genome sequencing.

What did you do? How did you do it? Please delete this section of merge section 2.4 and 2.6. If it is published elsewhere or in another manuscript, please add the information.

Ln134-142; 1000 ng is 1µg, uL is µL, umol/L is µmol L-1

Ln150; please add and reference to the LC30 value.

Section 3.2. Genome sequencing is completely without results. What did you do? How did you do it etc…? Please revise, delete or add a reference to the sequencing of the genome.

Ln214; what is “the sequenced PCR product”. Covering which part of the gene? Which exons? If you sequence “the PCR product” from “the cDNA pool of slected strains” you must get many different overlaying sequences. How do you analyse the data? It seems to be a wrong way to do it.

Ln218; you are making a reference to an “above mentioned transcriptome”. Which? Or did you make a RNAseq?

Ln219; how do you detect mutations from this pool? Please tell what you did.

Table 3; you do not need to show a table. You can write in Material and methods how many specimens were tested and in the Results section that the M1042 mutation was not found. Please delete Table 3.

Section 3.4. and Figure 3; it dose seem relevant with a phylogenetic study in the context of this resistance study. Additionally, the acquisition of the gene sequences are not described in this manuscript. Please delete Section 3.4. and Figure 3.

Section 3.5. and Figure 4; you present the “relative expression level”. What is your reference gene? How was the reference gene selected? How did you the make the ΔΔCt calculation?

Section 3.6., Figure 5; the expression level is very low compared to Figure 4. Why. What is your reference gene In this experiment?

Reviewer 5 Report

The manuscript entitled "Detection of chitin synthase mutations in lufenuron resistant Spodoptera frugiperda in China " written by Lv et al. is primarily interested in better understanding the role, if any, of chitin synthase in the fall armyworm following exposure to the insecticide lufenuron. In order to reach this objective, characterization of mutations in this gene target with relevance for lufenuron resistance as well as PCR-based quantification of transcripts coding for CHSA and CHSB in various tissues are presented. Results showcased here, albeit interesting to a certain extent, have limited scope for the field in general and several sections would require greater clarifications before this work would be deemed suitable for publication. Thus, this manuscript in its current form is not suitable for publication in Insects

Major

-          Abstract: This section can be markedly improved. More than one third of this section addresses conclusions (starting on line 17) for which data has not really been presented beforehand in that same section. Authors are strongly encouraged to list/state the specific results collected in this work as opposed to delving into general conclusions that are questionable in relevance for such a section or not supported by clear data (“we believe that…SfCHSA gene plays a greater role in the resistance of S. frugiperda against lufenuron.”).

-          Introduction: Lines 28-34 need editing. Several statements are unclear and cast doubt on the significance of the information relayed (“In the period of serious harm,…”; “…it was found invaded Yunnan…, etc.).

-          Introduction: Broad statements such as “IGRs will become one of the important pesticides for pest control all over the world because of its special mechanism of action and its advantages in sustainable agriculture” should be avoided in such a communication, unless supported by several examples from the literature that have highlighted the same conclusions as the authors. Please edit.  

-          Introduction: While the authors provide examples to illustrate a link between gene mutations and lufenuron resistance, examples showcasing the up-regulation of transcripts coding for chitin synthase, or of increased chitin synthase activity, in insects treated with this compound would add value to this work given that the authors do measure transcripts in insects treated with this chemical. 

-          Introduction: The statement “Studying which chitin synthase gene is more likely to be involved in insecticide resistance provides a basis for us to guide the scientific use of S. frugiperda” in its current state is unclear. It is not obvious, from the reader’s point of view, to comprehend how studying chitin synthase gene will “guide the scientific use of S. frugiperda”. This is one of many examples of statements throughout the manuscript that ought to be edited.

-          Section 2.4: A significant part of the results presented by the authors in this work resides on the data obtained following genome sequencing. Yet, this is the method that is the least described by the authors. Section 2.4 does not describe, in ways that an outsider could reproduce the work, what has been done by the research team to obtain the data presented. This is a major flaw.

-          Section 2.8: Please clarify the rationale, by using comparable studies for example, for selecting the concentration as well as the duration for the lufenuron exposures.    

-          Discussion: Given that the insect studied (fall armyworm) is amenable to dsRNA/RNAi work, this work would substantially benefit from RNAi-based approach that would strengthen the conclusions reached by the authors: “we believe that the insecticidal effect of larvae of lufenuron is most closely related to CHSA gene”. Such approach is an important step in studies such as this one to strengthen the link between over-expression of a given gene target and its role in the response of an insect to a given insecticide.

Minor

-          No need to use Spodoptera in bold on line 8.

-          No need to use lufenuron in bold on lines 10 and 11 (unless requested by journal).

-          Please define the acronym ”BPU” on line 45. It is defined later on, but should be defined the moment it first appears in the text.

-          Please define the acronym ”HDR” on line 47.

-          Please consider the term “Student's t -test”, if appropriate, on line 163.   

-          What is the rationale for a capital letter on “Insect” on line 48?

-          Lines 72-74 consist of a fragment and not a complete sentence.

-          Please address the single parenthesis on line 150.

-          Table 3 could probably be edited to be integrated as part of the main text. The need for such a table should be reconsidered.

-          Note for Figure 4 could be merged as part of the figure legend. Similar comment for Figure 5.

-          Please use “µ” as opposed to “u” when it is needed.

-          Consider referencing the article of Bolognesi et al., 2005, which notably measured levels of SfCHSB in the fall armyworm. It is strange that this study has not been cited here as it is greatly aligned with the presented study.

-           General: The authors are encouraged to seek advice/comments from within their network to improve the overall quality (written that is) of the manuscript as several minor errors can be identified throughout this document. Such guidance would also improve the overall clarity of several statements written to support this work.

Round 2

Reviewer 4 Report

Comments to the revised version of manuscript insects-1904599 intended as a research article in Insects entitled “Detection of chitin synthase mutations in lufenuron resistant Spodoptera frugiperda in China” by Lv et al. and six coauthors. In the revised manuscript the authors have commented and followed some suggestions by the reviewers.

Figure 1 is still not needed and should be replaced with a Supplementary Table with GPS coordinates or the GPS coordinates can be added to Table 2 or they could be mention as text in Section 2.1. This is a scientific publication presenting facts and not intuitively showing the sampling sites.

Table 3 is not needed; you did not find the M1040ATG mutation as neither hetero- nor homozygote and do not need to show that in a table. Please delete Table 3.

Figure 3 should be deleted, a phylogenetic study in the context of this resistance study is irrelevant. Please delete Section 3.3. and Figure 3.

Section 3.4: “Ribosomal protein L18 was used as a reference gene [39].” is better placed in Material and Methods section

Reviewer 5 Report

Comments have been addressed for the most part. This study would nevertheless benefit substantially from dsRNA/RNAi work as well as editing for clarity/conciseness. To consider. 

Revise these sentences for clarity:

(Line 48)", it was found invaded Yunnan province believed come from Myanmar"

(Line 150)"Compare the results of individual insect sequencing with sensitive populations to check if mutation I1040M (I1042M in P. xylostella ) have already occurred." 

(Line 313)Please clarify and support with a reference the recommandation: "we believe that resistance multiples below five times are low resistance, while higher than 10 times requires limited use."

Round 3

Reviewer 4 Report

The authors have followed comments and suggestions by reviewers and improved the manuscript.

Reviewer 5 Report

Authors have addressed the specific comments pertaining to the last revised version.